# The Effects of Viscosity and Capillarity on Nonequilibrium Distribution of Gas Bubbles in Swelling Liquid–Gas Solution

Alexander K. Shchekin *, Anatoly E. Kuchma and Elena V. Aksenova

Department of Statistical Physics, St. Petersburg State University, 7/9 Universitetskaya nab., 199034 St. Petersburg, Russia; an.kuchma@gmail.com (A.E.K.); e.aksenova@spbu.ru (E.V.A.)
* Correspondence: akshch@list.ru; Tel.: +7-911-949-56-66

**Abstract:** A detailed statistical description of the evolution of supersaturated-by-gas solution at degassing has been presented on the basis of finding the time-dependent distribution in radii of overcritical gas bubbles. The influence of solution viscosity and capillarity via internal pressure in the bubbles on this distribution has been considered until the moment when the gas supersaturation drops due to depletion and stops nucleation of new overcritical gas bubbles. This study is based on our previous results for the nonstationary growth rates of overcritical bubbles depending on gas supersaturation, diffusivity and solubility in solution, solution viscosity, and surface tension on bubble surface. Other important factors are linked with the initial rate of homogeneous gas bubble nucleation and coupling between diffusivity and viscosity in the solution. Here, we numerically studied how all these factors affect the time-dependent distribution function of overcritical bubbles in their radii, maximal and mean bubble radii, and the time-dependent swelling ratio of a supersaturated-by-gas solution in a wide range of solution viscosities.

**Keywords:** kinetics; degassing; gas bubble; diffusion; distribution function; capillarity; viscosity; swelling

## 1. Introduction

The theory of swelling of a gas-supersaturated solution as a result of nucleation and further growth of overcritical gas bubbles is in demand in many problems of fundamental science and related technologies. Such a theory is needed for describing bubble nucleation and subsequent foaming as a general phenomenon [1–3], the processes of intensive foaming in food and beverages [4–6], embolism in tissues and blood vessels [7–9], the formation of porous materials from polymeric melts [10–12], and volcanic eruptions [13–16]. The foundations of this theory were previously formulated in Refs. [17–20] and extended to the multicomponent case in Refs. [21,22]. In these works, the importance of the high solubility of the gas and its strong supersaturation in the initial solution was underlined as the conditions for the implementation of the mode of self-similar nonstationary diffusion growth of overcritical bubbles [17–19,23–25] with a significant swelling of the entire solution. It was also shown there that, in the case of stationary diffusion growth of gas bubbles, the swelling of the solution is insignificant and stays within a few percent. Further studies [26–29] showed that a marked role in the growth of gas bubbles can be played by capillary and viscous effects, which, through the pressure inside a bubble, slow down the transition to a self-similar regime and may even prevent it at all. More detailed analytical study of the limiting growth regimes and a numerical calculation of the growth rate of overcritical bubbles in the presence of capillary and viscous effects in a wide range of solution viscosities developed in Ref. [29] demonstrated, that even in the absence of the self-similar diffusion regime of bubble growth, the nonstationary gas concentration decline around the overcritical bubbles can be significant and provide nonstationary growth rates of bubbles [30]. It has also been mentioned [30] that coupling between diffusivity and

viscosity can enhance the damping effect on swelling with increasing solution viscosity. As a result of previous studies, we expect that the capillary and viscous effects may noticeably affect the general state of the supersaturated-by-gas solution and, in particular, its swelling at degassing. The first task of this paper is applying the analysis [29] to direct calculation of the distribution function of supercritical gas bubbles in a degassing solution at an arbitrary moment of the nucleation stage in the wide range of solution viscosities and nucleation rates with full allowance for the Laplace pressure in small overcritical bubbles. The novelty of this task lies in the fact that earlier, such calculations of the strong influence of the viscosity on the distribution function of the nucleated and growing gas bubbles with nonstationary diffusion were not carried out at all because of the lack of an analytical basis. Cooperation of the diffusivity and viscosity will be checked for the first time. The second task of this work is to study numerically the collective behavior of the ensemble of gas bubbles in degassing solution through the mean bubble radius and coefficient of swelling of the solution as functions of time and the parameters of the solution. Earlier, the collective characteristics of an ensemble of gas bubbles were considered only for the case of stationary diffusion in the absence of viscosity.

## 2. Materials and Methods

### 2.1. Concentration of Gas around a Single Growing Bubble

At nonstationary growth of a spherical gas bubble in a supersaturated-by-gas liquid solution, a concentric shell is formed around the growing bubble with a radially inhomogeneous distribution of the bulk gas concentration $c(r, t)$, where $r$ is the distance from the bubble center and $t$ is the time. The concentration $c(r, t)$ is described as a solution of the equation of nonstationary gas diffusion to the bubble with moving boundary,

$$\frac{\partial c(r, t)}{\partial t} = \frac{D}{r^2} \frac{\partial}{\partial r} \left[ r^2 \frac{\partial c(r, t)}{\partial r} \right] - \frac{R^2(t) \dot{R}(t)}{r^2} \frac{\partial c(r, t)}{\partial r} \tag{1}$$

where $D$ is the gas diffusivity in the liquid solution, $R$ is the bubble radius, and $\dot{R}(t)$ is the growth rate of the bubble radius. The boundary conditions for Equation (1) are

$$c(r, t) \underset{r \to \infty}{\to} c_0, \tag{2}$$

$$c(r = R, t) \equiv c(R) = c_\infty \left[ 1 + \frac{R_*}{R} \left( 1 + \frac{2\eta \dot{R}}{\sigma} \right) \right], \tag{3}$$

where $c_0$ is bulk concentration of the dissolved gas in the liquid solution, $c(R)$ is the equilibrium concentration of the dissolved gas at the bubble boundary, $c_\infty = c(R)|_{R \to \infty}$ is the equilibrium concentration of the dissolved gas in liquid solution at the flat boundary with the gas phase, $R_* \equiv 2\sigma / P_l$, $\sigma$ is the bubble surface tension, $P_l$ is the bulk pressure in the liquid solution, and $\eta$ is the viscosity of the liquid solution. Here, we have used Henry's law of solubility at the bubble surface with $c(R) = sn(R)$ ($s$ is the gas solubility in the liquid solution) and uniform concentration of gas in the bubble $n(R) = \frac{P_g}{k_B T} = \frac{P_l}{k_B T} \left( 1 + \frac{2\sigma}{P_l R} + 4\eta \frac{\dot{R}}{P_l R} \right)$ ($P_g$ is the gas pressure in the bubble, $k_B$ is the Boltzmann constant, and $T$ is the absolute temperature of solution, see details in Ref. [29]). In the case when contributions $R_*/R$ and $2\eta \dot{R} R_* / \sigma R$ cannot be neglected in Equation (3) (and this is just our case of interest with important capillary and viscous effects), there is an approximate solution of Equation (1) in the form [29,30]

$$c(r, t) = c(\rho, R) = c_\infty \left\{ \zeta_0 + 1 - \zeta_0 \left[ 1 - \frac{R_c}{R} \left( 1 + \frac{2\eta \dot{R}}{\sigma} \right) \right] \frac{\Phi\left(\rho, h(R)\frac{R\dot{R}}{2D}\right)}{\Phi\left(1, h(R)\frac{R\dot{R}}{D}\right)} \right\}, \tag{4}$$

where $\rho = r/R(t)$, $c_\infty = sP_l/k_BT$, and $\zeta_0 \equiv (c_0 - c_\infty)/c_\infty$ is the bulk gas supersaturation in the liquid solution and $R_c = R_*/\zeta_0$ is the critical radius of the bubble which is in unstable equilibrium with liquid solution at the bulk gas concentration $c_0$,

$$\Phi\left(\rho, h(R)\frac{R\dot{R}}{2D}\right) \equiv \int_\rho^\infty \frac{dx}{x^2} \exp\left[-\left(x^2 + \frac{2}{x} - 3\right)h(R)\frac{R\dot{R}}{2D}\right]. \tag{5}$$

Function $h(R)$ in Equation (5) is a correction function providing the balance of the number of gas molecules that have left the liquid solution and entered the growing bubble and is determined [29,30] as

$$h(R) = \frac{1 + \frac{2R_*}{3R} + \frac{4\eta\dot{R}}{3\sigma}\frac{R_*}{R} + \frac{2\eta R_*}{3\sigma}\frac{d\dot{R}}{dR}}{1 + \frac{R_*}{R} + \frac{2\eta\dot{R}}{\sigma}\frac{R_*}{R}} \tag{6}$$

while the growth rate $\dot{R}$ satisfies equation

$$\frac{h(R)R\dot{R}}{D} = s\zeta_0 \frac{1 - \frac{R_c}{R} - 2\frac{\eta\dot{R}}{\sigma}\frac{R_c}{R}}{1 + \frac{R_*}{R} + \frac{2\eta\dot{R}}{\sigma}\frac{R_*}{R}} \left(\Phi\left(1, h(R)\frac{R\dot{R}}{2D}\right)\right)^{-1}. \tag{7}$$

Thus, to determine $h(R)$ and $\dot{R}(R)$, it is necessary to solve jointly the system of Equations (6) and (7), then use Equation (5) to find the function $\Phi$ as a function of $\rho$ and $R$, and then substitute found function $\Phi$ into Equation (4) to calculate the gas concentration $c(\rho, R)$ in the solution around the single growing gas bubble.

### 2.2. Excluded Volume for a Single Growing Overcritical Bubble

Since the gas concentration $c(r, t)$ in the solution near the bubble, according to solution (4), drops to the equilibrium value, the local supersaturation $\zeta(r, t) = (c(r, t) - c_\infty)/c_\infty$ of the gas also drops there. It is convenient to determine the local relative decline $\varphi(r, t)$ of gas supersaturation from the bulk value $\zeta_0$ by the relation

$$\varphi(r, t) \equiv \frac{\zeta_0 - \zeta(r, t)}{\zeta_0}. \tag{8}$$

Taking into account Equation (4), we can rewrite Equation (8) as

$$\varphi(\rho, R) = \left(1 - \frac{R_c}{R}\left(1 + \frac{2\eta\dot{R}}{\sigma}\right)\right)\frac{\Phi\left(\rho, \frac{R\dot{R}}{2D}h(R)\right)}{\Phi\left(1, \frac{R\dot{R}}{2D}h(R)\right)}. \tag{9}$$

In the shell around the growing overcritical bubble, where the decline $\varphi(\rho, R)$ is maximal, the local nucleation rate $I(\rho, R)$ of new overcritical bubbles is suppressed. At the same time, outside this shell, the nucleation rate increases to the initial level $I_0$. According to Refs. [17–20,29], it is convenient to introduce the volume $V_{ex}(R)$, which is excluded for nucleation around the selected growing overcritical bubble with radius $R(t)$ by the integral relation:

$$V_{ex}(R) = 4\pi R^3 \int_1^\infty d\rho \rho^2 \left(1 - \frac{I(\rho, R)}{I_0}\right). \tag{10}$$

According to Equation (10), the total number of bubbles formed per unit time at current profile of gas supersaturation in the solution volume $V$ around a selected bubble is equal to the number of bubbles nucleated in the volume $V - V_{ex}(R)$ where the nucleation rate is assumed to be equal to $I_0$ at bulk gas supersaturation $\zeta_0$. The ratio $q(R) \equiv V_{ex}(R)/V_R$

of volume $V_{\text{ex}}(R)$ and volume $V_R = (4\pi/3)R^3$ of the bubble with radius $R$ becomes an important parameter. Figure 1 illustrates the physical meaning of the quantities $V_{\text{ex}}$ and $q$.

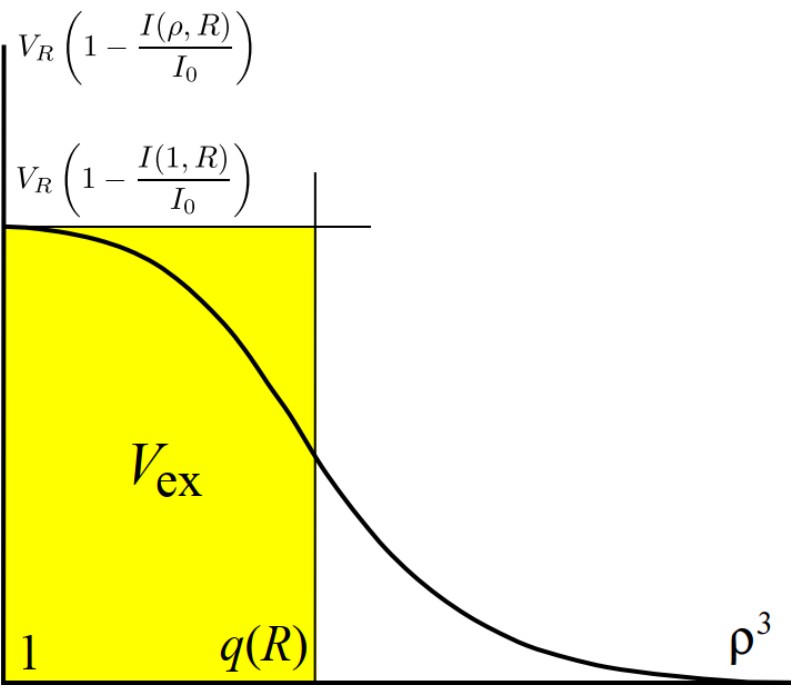

**Figure 1.** The relation between $V_{\text{ex}}(R)$, $V_R$, and $q(R)$.

As is generally known [17–20], the ratio $I(\rho, R)/I_0$ can be expressed through the decline $\varphi(\rho, R)$ as $I(\rho, R)/I_0 = e^{-\Gamma\varphi(\rho,R)}$, where $\Gamma \equiv -\zeta_0(\partial \Delta F_c/\partial \zeta_0)$ and $\Delta F_c$ is the activation barrier for formation of the critical gas bubble. With the help of Equation (10), we can represent ratio $q$ as

$$q(R) = 3\int_1^\infty d\rho \rho^2 \left(1 - e^{-\Gamma\varphi(\rho,R)}\right). \tag{11}$$

As we will see in the next section, this parameter plays an important role in finding the total excluded volume for an ensemble of growing bubbles.

### 2.3. Total Excluded Volume and Distribution of Overcritical Gas Bubbles in Radii

We assume that the dissolved gas at degassing only redistributes between the volume of the solution and the nucleated and growing bubbles. Let us now designate $V_{\text{ex}}^{\text{tot}}(t)$ as the sum of individual excluded volumes around all bubbles at time $t$. If the initial volume of solution equals $V_L$, then $V_L - V_{\text{ex}}^{\text{tot}}(t)$ is the solution volume, where initial nucleation rate $I_0$ remains preserved to time $t$.

$$f(R,t)V_L = I_0 \int_0^t d\tau \left[V_L - V_{\text{ex}}^{\text{tot}}(\tau)\right]\delta(R - R(t - \tau)), \tag{12}$$

where the Dirac delta-function $\delta(R - R(t - \tau))$ takes into account the spread in radii of the bubbles past the critical size at previous moments of time. Evidently, the total volume $V_g(t)$ of bubbles at time $t$ is related to the distribution function $f(R,t)$ as

$$V_g(t) = \frac{4\pi V_L}{3} \int_{R_0}^\infty R^3 f(R,t)dR, \tag{13}$$

where $R_0$ is the radius of the smallest overcritical bubble. Because Equation (11) for the excluded volume ratio $q(R) \equiv V_{ex}(R)/V_R$ can be applied to any overcritical bubble, we assume [29] that rate $dV_{ex}^{tot}(t)/dt$ is proportional to integral $\frac{4\pi V_L}{3}\int_{R_0}^{\infty} dR q(R)R^3 \frac{\partial f(R,t)}{\partial t}$ (which is a direct change in time of the total excluded volume) and, in view of the possibility of an accidental overlapping of diffuse layers of bubbles at any point in a liquid solution, is also proportional to the fraction $1 - V_{ex}^{tot}(t)/V_L$ of initial liquid solution volume which is available for the nucleation of new bubbles. Thus, we may write

$$\frac{dV_{ex}^{tot}(t)}{dt} = \frac{4\pi V_L}{3}\left(1 - \frac{V_{ex}^{tot}(t)}{V_L}\right)\int_{R_0}^{\infty} q(R)R^3 \frac{\partial f(R,t)}{\partial t} dR. \tag{14}$$

As shown in Ref. [29], with the help of Equation (12), Equation (14) can be converted to equation

$$\frac{V_L - V_{ex}^{tot}(t)}{V_L} = \exp\left(-I_0 \int_0^t d\tau q(R(t-\tau))V_R(t-\tau)\frac{V_L - V_{ex}^{tot}(\tau)}{V_L}\right). \tag{15}$$

For the nucleation stage, the first iteration solution of Equation (15) may be written as

$$1 - \frac{V_{ex}^{tot}(t)}{V_L} \approx \exp\left[-\frac{4\pi I_0}{3}\int_{R_0}^{R(t)} \frac{q(R)R^3}{\dot{R}(R)} dR\right]. \tag{16}$$

Substitution of Equation (16) into Equation (12) gives

$$f(R,t) \approx \frac{I_0 \theta\left(t - \widetilde{t}(R)\right)}{\dot{R}(R)} e^{-\frac{4\pi I_0}{3}\int_{R_0}^{R(t-\widetilde{t}(R))} \frac{q(y)y^3}{\dot{R}(y)} dy}, \tag{17}$$

where $\theta$ is the Heaviside step function and $\widetilde{t}(R)$ is the time of overcritical bubble growth from the initial radius $R_0$ to radius $R$ (the inverse function for $R(t)$), $R(0) = R_0$. With the help of Equation (17) we can find the mean bubble radius $\overline{R}(t)$:

$$\overline{R}(t) = \frac{\int_{R_0}^{\infty} R f(R,t) dR}{\int_{R_0}^{\infty} f(R,t) dR} = \frac{\int_{R_0}^{R(t)} \frac{R dR}{\dot{R}(R)} e^{-\frac{4\pi I_0}{3}\int_{R_0}^{R(t-\widetilde{t}(R))} \frac{q(y)y^3}{\dot{R}(y)} dy}}{\int_{R_0}^{R(t)} \frac{dR}{\dot{R}(R)} e^{-\frac{4\pi I_0}{3}\int_{R_0}^{R(t-\widetilde{t}(R))} \frac{q(y)y^3}{\dot{R}(y)} dy}}. \tag{18}$$

As follows from Equation (16), the duration $t_1$ of the nucleation stage is determined by the condition

$$\frac{4\pi I_0}{3}\int_{R_0}^{R(t_1)} \frac{q(R)R^3}{\dot{R}(R)} dR = 1. \tag{19}$$

where $R(t_1)$ is the maximal bubble radius to the end of the nucleation stage.

### 2.4. Swelling of Solution

One of the important collective characteristics of the state of the degassing solution is its swelling over time. Assuming that the dissolved gas does not leave the solution through its external boundaries, we can determine the swelling coefficient $\kappa$ as

$$\kappa(t) \equiv \frac{V_L + V_g(t)}{V_L}. \tag{20}$$

Substitution of Equations (13) and (17) into Equation (20) gives the following expression for swelling coefficient $\kappa$ as a function of time $t$:

$$\kappa(t) = 1 + \frac{4\pi I_0}{3} \int\limits_{R_0}^{R(t)} \frac{dR R^3}{\dot{R}(R)} e^{-\frac{4\pi I_0}{3} \int\limits_{R_0}^{R(t-\tilde{t}(R))} \frac{q(y)y^3}{\dot{R}(y)} dy}. \tag{21}$$

### 3. Results and Discussion

According to the tasks formulated in the Introduction, we are interested in direct calculation of the distribution function $f(R,t)$ of supercritical gas bubbles in a degassing solution and finding the mean bubble radius $\overline{R}(t)$ and coefficient $\kappa(t)$ of swelling of the solution at an arbitrary time moment of the nucleation stage in the wide range of solution viscosities and nucleation rates with full allowance for the Laplace pressure in small overcritical bubbles. As is seen from Equations (17) and (21), the quantities $f(R,t)$ and $\kappa(t)$ are determined by the dependence of the individual bubble radius $R(t)$ on time (related to the bubble growth rate $\dot{R}(R)$) and the dependence of the excluded volume ratio $q(R)$ (related to the local relative decline $\varphi(r,R,t)$ of gas supersaturation $\zeta(r,R,t)$ around the bubble with radius $R$ from the bulk supersaturation value $\zeta_0$). To illustrate the predictions of the theory considered in Section 2 at different values of solution viscosity $\eta$ and different nucleation rates $I_0$, we carried out several numerical calculations. First, we substituted Equation (6) in Equation (7) and have used Maple (Maplesoft, a division of Waterloo Maple Inc. 2021) implicitplot and getdata routines for finding $R\dot{R}/D$ as a function of bubble radius $R$ using 1500 points. Numerical integration of the obtained $R\dot{R}(R)$- dependence gave the function $R(t)$. Substitution of the $R\dot{R}/D(R)$ and $\dot{R}(R)$ functions in Equation (6) provided the correction function $h(R)$. Then, with the help of Equation (9), we found the local relative decline $\varphi(r,t)$ of gas supersaturation around a single bubble in variables $\rho$ and $R$, and substituted the results in Equation (12) to calculate the excluded volume ratio $q$ as a function of the bubble radius $R$. As a next step, we used the results for $\dot{R}(R)$ and $q(R)$ in Equations (17) and (18) to calculate the distribution function $f(R,t)$ of overcritical bubbles and the dependence of the mean bubble radius $\overline{R}(t)$ on time. Equation (19) allows us to find the duration $t_1$ of the nucleation stage. Finally, with the help of Equation (20) we computed the dependence of the swelling coefficient $\kappa$ on time $t$.

The following values of parameters for the solution and the dissolved gas have been taken in calculation:

$$P_l = 10^5 \text{ Pa}, \ \sigma = 0.075 \text{ N/m}, \ s = 0.2, \ \zeta_0 = 50, \ \Gamma = 50, \ D = 10^{-9} \text{ m}^2/\text{s}, \tag{22}$$

that gave us

$$R_* = 2\sigma/P_l = 1.5 \cdot 10^{-6} \text{m}, \ R_c = R_*/\zeta_0 = 3 \cdot 10^{-8} \text{ m}. \tag{23}$$

Evidently, the parameters in (22) can have other values that are acceptable from a physical point of view. Choosing values (22), we did not pursue the goal of describing any particular system but only illustrating the role of viscosity in a wide range of its values. The accepted values of the parameters are close to those that can be appropriate for solutions of highly soluble gases in water.

Generally, the initial nucleation rate $I_0$ is the function of initial supersaturation $\zeta_0$ [1,22] which is mostly characterized by derivative $\Gamma \equiv -\zeta_0(\partial \Delta F_c/\partial \zeta_0)$ (mentioned in Section 2). Pre-exponential factor in the relation between $I_0$ and activation barrier $\Delta F_c$ can give a second-order correction. Taking in mind the discussion in [31] concerning the relation between the activation barrier $\Delta F_c$ for the classical theory of homogeneous bubble gas nucleation in gas-supersaturated liquid solution, the gas solubility, and supersaturation, we decided to take two values for the initial nucleation rate $I_0$: $I_0 = 10^5$ m$^{-3}$s$^{-1}$ and $I_0 = 10^7$ m$^{-3}$s$^{-1}$ at the same values of $\zeta_0$ and $\Gamma$ fixed in Equation (22). The influence of the solution viscosity $\eta$ has been checked at three values of viscosity: $\eta = 1$ Pa·s, $\eta = 10^3$ Pa·s, and $\eta = 10^5$ Pa·s.

Note that it is possible to study the degassing of various gases in a solvent, and this allows us to formally consider the case when the diffusion coefficient is fixed at different values of the viscosity of the solution. If we study the degassing of a given gas in solutions with different viscosities, then we should take into account the relationship between the diffusivity and viscosity. In such a case, we have used in our calculations the Einstein-Stokes formula which predicts that the product of the diffusion coefficient for gas molecules and viscosity of solution is constant at other fixed parameters.

The results of the computations are presented below. The dependence of an individual bubble radius $R(t)$ on time $t$ is shown in Figure 2 for three values of the solution viscosity. The dashed blue curve 0 depicts the limiting case at $\eta = 0$ and $R \gg R_*$ which corresponds to self-similar growth of bubbles [17,19,29]. In this limiting case, all viscosity and capillary effects are absent. Solid curves 1–3 correspond at fixed diffusivity $D = 10^{-9}$ m$^2$/s to $\eta = 1$ Pa·s, $\eta = 10^3$ Pa·s, and $\eta = 10^5$ Pa·s, respectively. Curve 2′ is plotted for the solution with $\eta = 10^3$ Pa·s, but the product $D\eta$ was taken to be the same as for the solution with $D = 10^{-9}$ m$^2$/s and $\eta = 1$ Pa·s. It is seen that increasing the viscosity at fixed diffusivity decreases the bubble growth rate and diminishes $R(t)$. However, this effect is significant only for curves 3 and 2′. Evidently, nucleation rate does not affect $R(t)$.

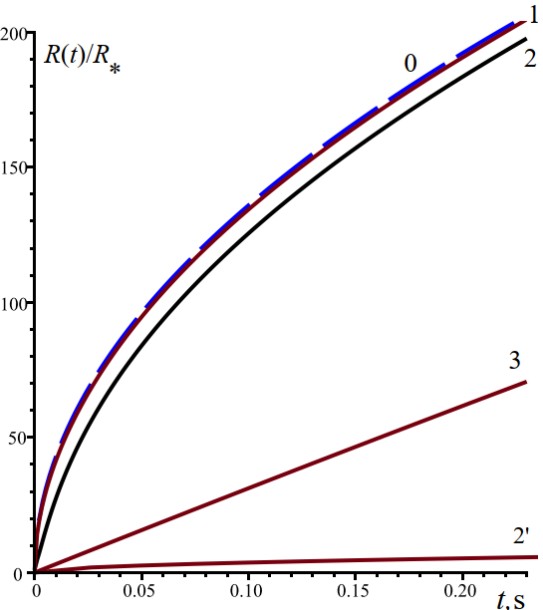

**Figure 2.** Bubble radius $R(t)$ as a function of time at fixed diffusivity and three values of viscosity.

Figure 3a,b illustrate how the behavior of the local relative decline $\varphi(r, R(t))$ of gas supersaturation changes with increasing viscosity at fixed diffusivity. If we neglect the capillarity and viscosity, the gas concentration near the bubble is equal to the equilibrium value $c_\infty$ and, respectively, the local supersaturation of the gas near the bubble at $r = R(t)$ is zero. Then, the local relative decline $\varphi(r = R(t), R(t))$ is equal to unity for all bubble sizes. As the distance $r$ from the bubble center increases, the value of $\varphi(r, R(t))$ drops

to zero. This pattern corresponds to the behavior of $\varphi(r, R(t))$ on surface 0 in Figure 3a, and we see that the decay of $\varphi(r, R(t))$ is quite fast for the values of the parameters taken from Equation (22) (nucleation rate does not affect $\varphi(r, R(t))$). It is also seen that rising viscosity leads to a slower decrease in the value of $\varphi(r, R(t))$ in the space around the bubble, especially for smaller bubble sizes. The presence of folds on surfaces 1 and 2 in the region of small droplet radii in Figure 3a is just due to viscous and capillary effects. Compared with the dependence of the radius on time shown in Figure 2, the influence of these effects on $\varphi(r, R(t))$, even at a fixed diffusion coefficient, is noticeably more significant.

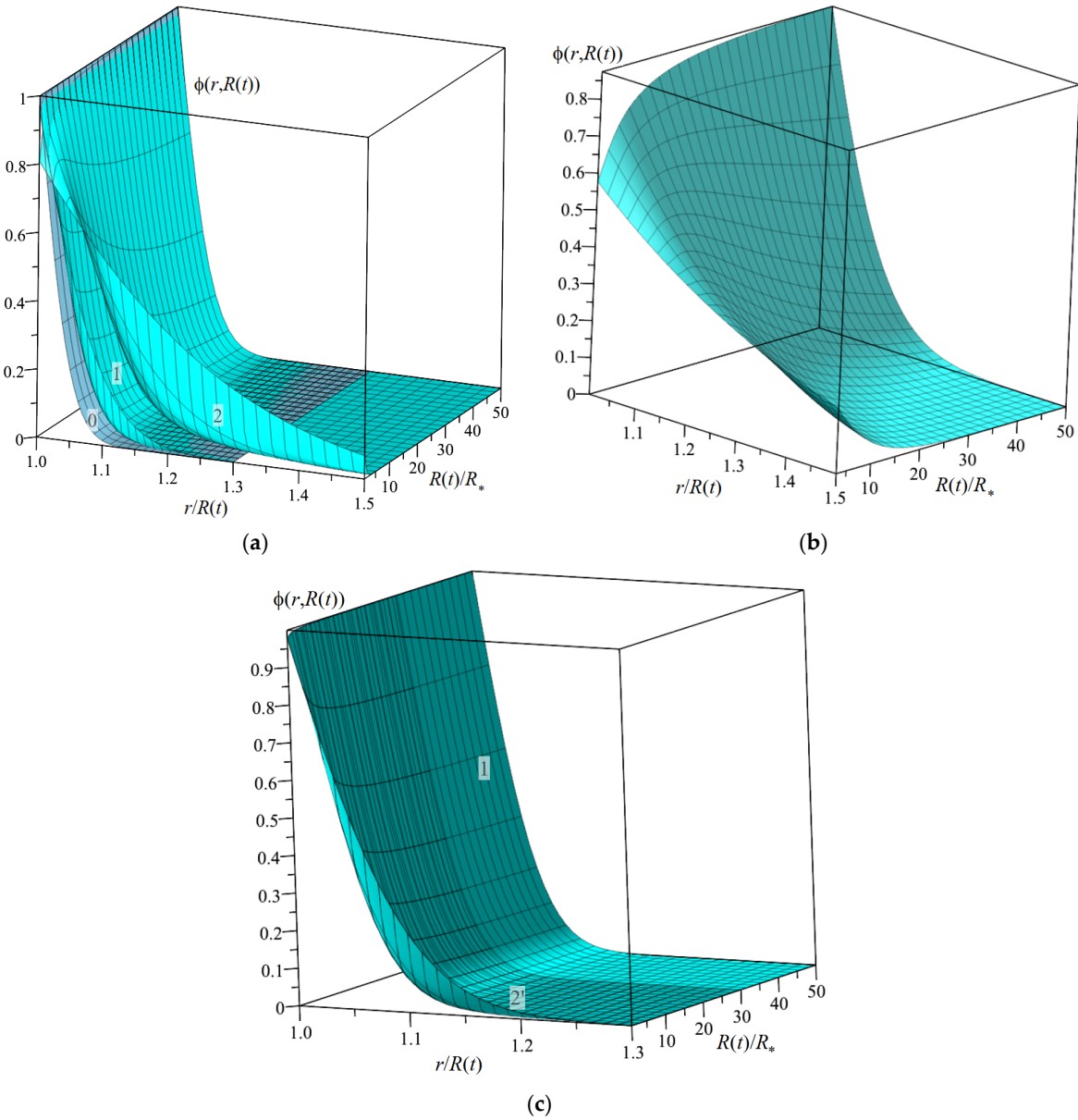

**Figure 3.** (**a**) The 3D-surfaces of local relative decline $\varphi(r, R(t))$ of gas supersaturation at distance $r$ around the bubble with radius $R(t)$. Surface 0 plots the limiting case at $\eta = 0$, $D = 10^{-9}$ m²/s, and $R \gg R_*$. Surfaces 1 and 2 correspond to $\eta = 1$ Pa·s and $\eta = 10^3$ Pa·s at $D = 10^{-9}$ m²/s. (**b**) The 3D-surface of local relative decline $\varphi(r, R(t))$ of gas supersaturation at distance $r$ around the bubble with radius $R(t)$ at $\eta = 10^5$ Pa·s and $D = 10^{-9}$ m²/s. (**c**) The 3D-surfaces of local relative decline $\varphi(r, R(t))$ of gas supersaturation at distance $r$ around the bubble with radius $R(t)$ for solutions with $\eta = 1$ Pa·s, $D = 10^{-9}$ m²/s (surface 1) and $\eta = 10^3$ Pa·s, $D = 10^{-12}$ m²/s (surface 2′).

As is seen in Figure 3b, viscosity has much a stronger influence on the value of $\varphi(r, R(t))$ at $\eta = 10^5$ Pa·s. However, we can observe here that $\varphi(r, R(t))$ still monotonically drops along the lines with fixed $R(t)$ as in Figure 3a, and this fact is in accordance with the general behavior of the solution of the diffusion equation. The nonmonotonic behavior of function $\varphi(r, R(t))$ as a function of the bubble radius $R(t)$ for small radii at a fixed distance $r/R(t)$ from the bubble center, which is well observed in Figure 3b (and weakly noticeable in Figure 3a), is associated with the effect of viscosity on the bubble growth rate $\dot{R}$ in the factor $\left(1 - (R_c/R)\left(1 + 2\eta\dot{R}/\sigma\right)\right)$ in the expression (9).

We also compared surfaces of $\varphi(r, R(t))$ for solutions with fixed product $D\eta$ at $\eta = 1$ Pa·s, $D = 10^{-9}$ m²/s and $\eta = 10^3$ Pa·s at $D = 10^{-12}$ m²/s (i.e., in the case of coupling of diffusivity and viscosity). The result is shown in Figure 3c. As is seen, the functions $\varphi(r, R(t))$ in scaled variables $r/R(t)$ and $R(t)/R_*$ virtually coincide in this case. This is not a random result. A study of the asymptotical forms of the growth rate for small and large $R$, carried out in [30], showed that they are directly proportional to the diffusion coefficient, and only the terms proportional to the product $D\eta$ are present in the corrections. Our calculations here additionally show that the whole curves for $\dot{R}(R)/D$ and, in view of Equation (6), for correction function $h(R)$ coincide at different viscosities and diffusion coefficients with a fixed product $D\eta$.

The next important characteristic of gas absorption by a bubble is the excluded volume ratio $q$. The dependence of the ratio $q$ on bubble radius $R(t)$ at fixed diffusivity $D = 10^{-9}$ m²/s and three values of viscosity $\eta = 1$ Pa·s, $\eta = 10^3$ Pa·s, and $\eta = 10^5$ Pa·s is shown in Figure 4 (curves 1, 2, and 3, respectively). Curve 2′ is related to the solution with $\eta = 10^3$ Pa·s and $D = 10^{-12}$ m²/s. As is outlined by the blue color, this curve virtually coincides with the curve 1 for the solution with $\eta = 1$ Pa·s and $D = 10^{-9}$ m²/s (with the same product $D\eta$). As follows from definition (11), this coincidence is a consequence of the coincidence of the surfaces $\varphi(r, R(t))$ for curves 1 and 2′. The value $q_0 = 0.3399$, which is shown by line 0, corresponds to self-similar growth of bubbles in the limiting case at $\eta = 0$ and $R \gg R_*$ at the diffusivity $D = 10^{-9}$ m²/s with neglection of all viscosity and capillary corrections. This value is asymptotic for other curves. Increasing the viscosity at fixed diffusivity increases the bubble size to reach this asymptotic value.

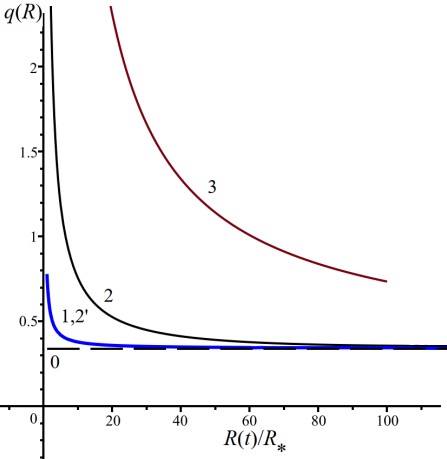

**Figure 4.** The excluded volume ratio $q$ as a function of the bubble radius $R$ at different conditions on diffusivity and viscosity.

Figure 5a,b show 3D plots of the nonequilibrium distribution function $f(R, t)$ as a function of the bubble radius and time for different values of viscosity with fixed diffusion coefficient $D = 10^{-9}$ m²/s and nucleation rate $I_0 = 10^5$ m⁻³s⁻¹. Figure 5a illustrates the behavior of $f(R, t)$ at $\eta = 1$ Pa·s (surface 1) and $\eta = 10^3$ Pa·s (surface 2) relative to the distribution function for the limiting case at $\eta = 0$ and $R \gg R_*$ (surface 0) with neglection of viscosity and capillarity effects. We used the semi-transparency of the plotted surfaces.

Therefore, it is clear that under the specified parameters, all distribution functions are similar and successively nested into each other. Figure 5b shows the behavior of $f(R, t)$ at $\eta = 10^5$ Pa·s. We see that in all cases considered, in accordance with Formula (17), the presence of the Heaviside theta function leads to the appearance of a front, beyond which the distribution function is equal to zero. At any moment of time $t$ there cannot be bubbles with radii greater than $R(t)$. At the distribution front, the function $f(R, t)$ reaches its maximum values for each bubble radius $R$. A more specific behavior of $f(R, t)$ at small bubble radii in Figure 5b is related to the fact that the bubble growth rate $\dot{R}(t)$ is very small for these sizes (see analytical results in [30]).

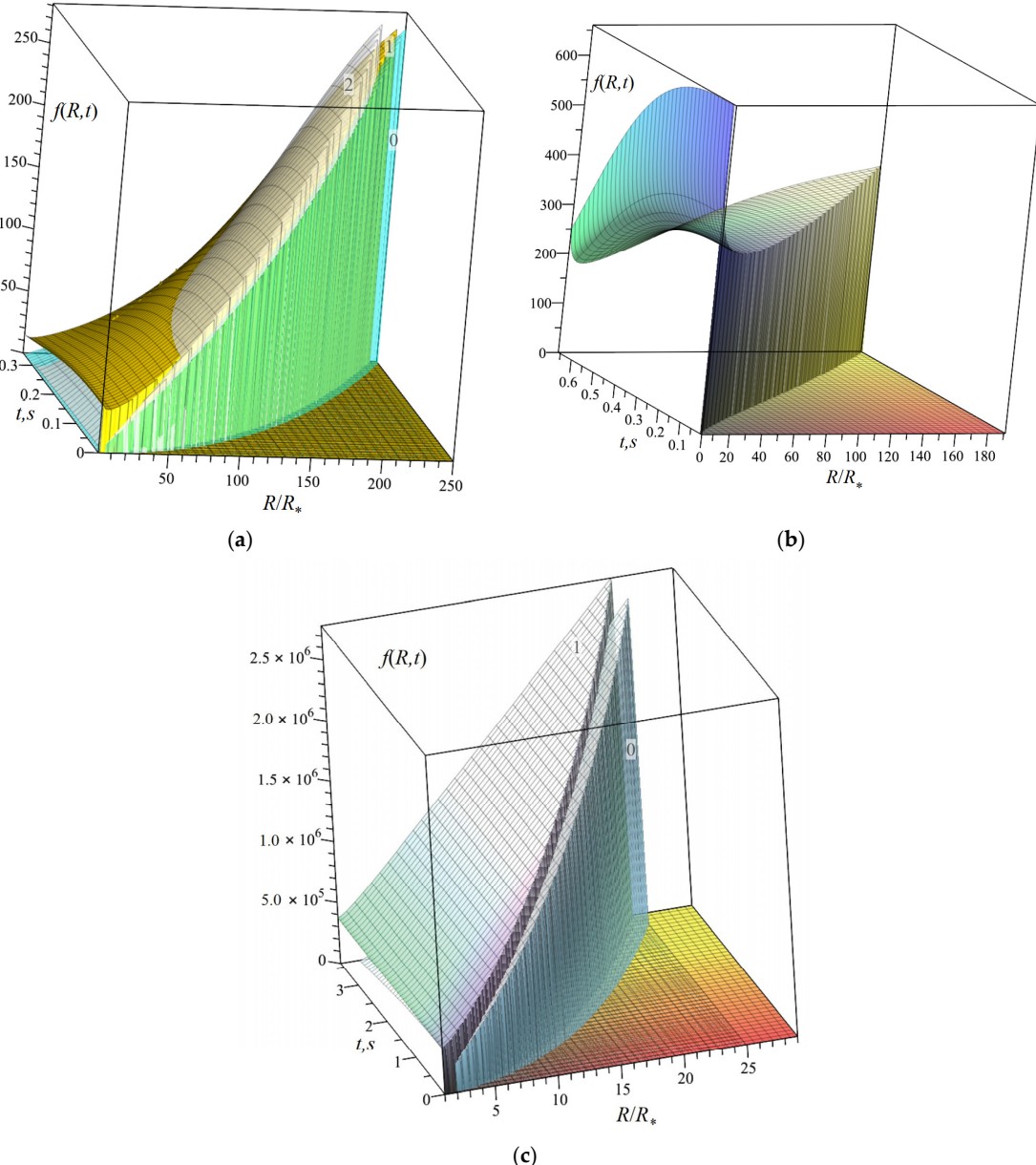

**Figure 5.** (**a**) The 3D-surfaces of the gas bubble distribution $f(R, t)$ as a function of the bubble radii $R$ and time $t$. Surface 0 plots the limiting case at $\eta = 0$ and $D = 10^{-9}$ m²/s. Surfaces 1 and 2 corresponds to $\eta = 1$ Pa·s and $\eta = 10^3$ Pa·s at $D = 10^{-9}$ m²/s. (**b**) The 3D-surface of the gas bubble distribution $f(R, t)$ as a function of the bubble radii $R$ and time $t$ at $\eta = 10^5$ Pa·s and $D = 10^{-9}$ m²/s. (**c**) The 3D-surfaces of the gas bubble distribution $f(R, t)$ as a function of the bubble radii $R$ and time $t$ at $D = 10^{-12}$ m²/s. Surface 0 plots the limiting case at $\eta = 0$ and $R >> R_*$. Surfaces 1 corresponds to $\eta = 10^3$ Pa·s.

Of particular interest is the behavior of the distribution function of gas bubbles at higher nucleation rates and when the diffusion coefficient of gas molecules and the viscosity of the solution are coupled. The results of the corresponding calculations of $f(R, t)$ for solution with $\eta = 10^3$ Pa·s and $D = 10^{-12}$ m²/s are shown in Figure 5c in relation to the distribution function for the case with neglection of viscosity and capillarity effects at $D = 10^{-12}$ m²/s.

Note that the maximal value of time $t$ in Figure 5a–c corresponds to the duration of the nucleation stage during solution degassing. According to Equation (19), the characteristic time $t_1$ of the nucleation stage at parameters (22) and nucleation rate $I_0 = 10^5$ m⁻³s⁻¹ is 0.30 s for a solution with at $\eta = 1$ Pa·s (the maximal bubble radius is $241 R_*$), 0.34 s for a solution with $\eta = 10^3$ Pa·s (the maximal bubble radius is $244 R_*$), and 0.66 s for a solution with $\eta = 10^5$ Pa·s (maximal bubble radius is $191 R_*$). At $D = 10^{-12}$ m²/s and $\eta = 10^3$ Pa·s, we find $t_1 = 3.62$ s and maximal bubble radius is $R(t_1) = 24 R_*$. In comparison with surface 2 in Figure 5a related to $I_0 = 10^5$ m⁻³s⁻¹, we see in Figure 5c a sharp rise in the number of small bubbles with a significant increase in the degassing time.

Figure 6 shows the dependence of the mean bubble radius $\overline{R}(t)$ on time at the nucleation stage, calculated by Formula (18). The numbering of the curves (with used values of physical parameters) in Figure 6 corresponds to the numbering of the curves in Figure 2 for the dependence of the bubble radius on time $t$. Obviously, the curves in Figure 2 can be interpreted as dependences of the maximal bubble size in the ensemble of bubbles.

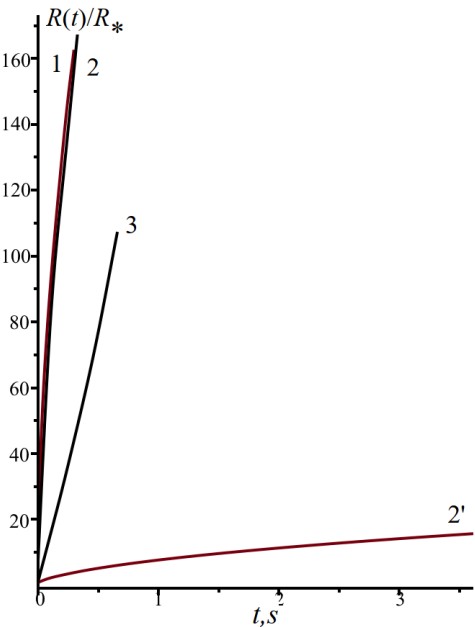

**Figure 6.** Mean bubble radius $\overline{R}(t)$ as a function of time at different conditions on diffusivity and viscosity.

Comparing the curves in Figure 6 with the corresponding curves in Figure 2, we conclude that the average radius in an ensemble of bubbles always grows noticeably slower than the maximum bubble radius. An increase in viscosity slows down the growth of the mean radius $\overline{R}(t)$, especially in the case of inverse proportionality between the diffusion coefficient and the viscosity of the solution. The maximal value of time $t$ for curves in Figure 6 corresponds to the duration $t_1$ of the nucleation stage during solution degassing.

Finally, Figure 7 shows the time dependence of the swelling coefficient of the degassing solution during the nucleation stage. The numbering of the curves under different degassing conditions coincides with the numbering in Figure 6. It should be noted that at all three values of viscosity (fixed and inversely proportional to the viscosity diffusivity),

swelling of the solution at the nucleation stage rises approximately to close values, although at a significantly different rate.

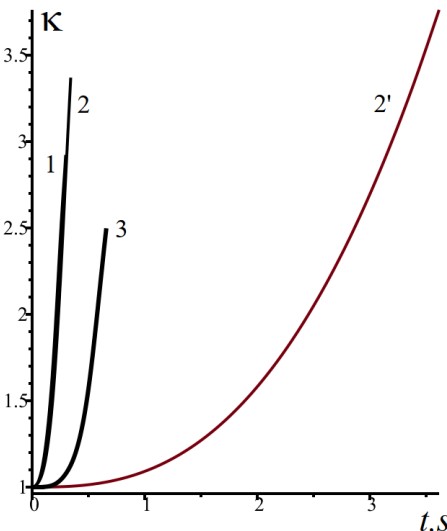

**Figure 7.** Swelling coefficient $\kappa(t)$ for the degassing solution as a function of time during the nucleation stage at different conditions on diffusivity and viscosity.

The fact that the swelling coefficient of curve 2′ with inverse proportionality between the diffusion coefficient and viscosity reaches greater values by the end of the nucleation stage than the swelling coefficient of curve 2 at a fixed diffusion coefficient is due to the use of a higher nucleation rate in the calculations for the case 2′. As a result, we have in the case of 2′ a huge number of small bubbles to the end of nucleation stage.

## 4. Conclusions

Our previous analysis [29,30] of individual growth rates of gas bubbles at degassing in viscous supersaturated-by-gas solutions under high initial gas supersaturation and gas solubility has been extended to study of cooperative effects for an ensemble of growing gas bubbles. As we have shown here, viscous effects may noticeably affect the distribution function of supercritical gas bubbles in a degassing solution on the nucleation stage, and we gave a description of the mechanism of a such influence. In particular, as a macroscopic effect at degassing, viscosity provides slowdown of the maximal and mean bubble growth as well as the rate of swelling of the total solution. With increasing solution viscosity by five decimal orders at fixed diffusivity, viscosity does provide the damping effect on the rate of growth of the mean bubble radius and the rate of swelling up to two times and, correspondingly, increases twice the time of the nucleation stage. Taking inverse coupling between diffusivity and viscosity into account allowed us to find much stronger dumping up to one decimal order for three decimal orders of viscosity. Nevertheless, we have shown that to the end of nucleation stage, the swelling coefficient will be several units. Thus, in spite of the fact that the self-similar theory of diffusional growth of bubble [17–19] (which requires $\eta = 0$ and $R >> R_*$) cannot be directly applied at large viscosities, many of its features are preserved.

The results obtained allow us to consider any specific system for which the physicochemical parameters, such as gas solubility, surface tension, diffusion and viscosity coefficients, nucleation rate, and work of formation are self-consistent. At present, such a set of values of parameters can only be obtained experimentally.

In this paper, we considered the case of instantaneous decompression of a gas-saturated solution. In the case of a finite decompression rate, the approach proposed here requires modification, primarily related to the change in the growth dynamics of a

single gas bubble. The corresponding problem of single bubble growth in a highly viscous liquid was considered in recently published paper by Chernov, Davydov, and Pil'nik [32].

**Author Contributions:** Conceptualization, Writing—Original Draft Preparation, Computations, A.K.S.; Conceptualization, Validation, Formal analysis, A.E.K.; Review and Editing, Methodology, Validation, Visualization, E.V.A. All authors have read and agreed to the published version of the manuscript.

**Funding:** This work was supported by the Russian Science Foundation under grant no. 22-13-00151, https://rscf.ru/en/project/22-13-00151/ (accessed on 30 April 2023).

**Institutional Review Board Statement:** Not applicable.

**Informed Consent Statement:** Not applicable.

**Data Availability Statement:** Not applicable.

**Conflicts of Interest:** The authors declare no conflict of interest.

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
