# Peer review of "The Effects of Viscosity and Capillarity on Nonequilibrium Distribution of Gas Bubbles in Swelling Liquid–Gas Solution"

_colloids, doi:10.3390/colloids7020039_

Round 1

Reviewer 1 Report

This work builds upon two recent publications by the authors in JCP and Physica A. Here they extended their previous analysis to calculate the distribution function of supercritical gas bubbles in a degassing solution and the swelling coefficient of the solution. Therefore, the fundamental scientific advancement in this work appears to be limited.

General comments:

1.     The relationship between the current work and the authors' previous works should be discussed in more detail, considering that this work is an extension of their previous research.

2.     The authors should provide validations for their theoretical predictions and establish possible connections to experimental observations, which would enhance the credibility of their predictions.

3.     How are the values of the parameters in Eq. (22) determined?

Author Response

We thank the referee for his work and valuable comments. Following these comments, we revised the manuscript. Here is our point-by-point response to the reviewer's main three comments, along with the corresponding changes to the text.

1. The relationship between the current work and the authors' previous works should be discussed in more detail, considering that this work is an extension of their previous research.

The relation between the this paper and our previous works are considered in Introduction. We added the following statements in Introduction on lines 54-58 ...The novelty of this task lies in the fact that earlier such calculations of the strong influence of the viscosity on the distribution function of the nucleated and growing gas bubbles with non-stationary diffusion were not carried out at all because of the lack of an analytical basis. Cooperation of the diffusivity and viscosity will be checked for the first time... and on lines 61-63 ...Earlier, the collective characteristics of an ensemble of gas bubbles were considered only for the case of stationary diffusion in the absence of viscosity.

2. The authors should provide validations for their theoretical predictions and establish possible connections to experimental observations, which would enhance the credibility of their predictions.

We considered the validity of our theoretical predictions in Conclusions. All the results may be experimentally checked. To underline this possibility and the difficulties on its realization, we added on lines 364-372 ...

The results obtained allow us to consider any specific system for which the physicochemical parameters, such as gas solubility, surface tension, diffusion and viscosity coefficients, nucleation rate, and work of formation are self-consistent. At present, such a set of values of parameters can only be obtained experimentally.

In this paper, we considered the case of instantaneous decompression of a gas-saturated solution. In the case of a finite decompression rate, the approach proposed here requires modification, primarily related to the change in the growth dynamics of a single gas bubble. The corresponding problem of single bubble growth in a highly viscous liquid was considered in recently published paper by Chernov, Davydov, and Pilnik [32]... 

3. How are the values of the parameters in Eq. (22) determined?

We commented the choice of the parameters in eq.(22) in lines 209-213...Evidently, the parameters in (22) can have other values that are acceptable from a physical point of view. Choosing values (22), we did not pursue the goal of describing any particular system, but only illustrating the role of viscosity in a wide range of its values. The accepted values of the parameters are close to those that can be appropriate for solutions of highly soluble gases in water...

Reviewer 2 Report

The authors gave an analytical expression for bubble growth in supersaturated solution. Although the results seems interesting, they did not give a clear presentation for their results, particularly for the derivation of equations. For example, they gave equation (1) directly with no explanation for the underlying physics. I understand that the equations may be followed from previous publications. But the authors should also give a clear and explicit for the underling physics. 

Moreover, there are many results and theories for bubble nucleation in statistical thermodynamics and molecular simulations. The authors seem rarely mentioned those results. They should comment and compare their theories and results with the relevant references. 

With considering these points, I can not recommend the publication of this manuscript at the present form. 

Author Response

We thank the referee for his work and valuable comments. Following these comments, we revised the manuscript. Here is our point-by-point response to the reviewer's comments, along with the corresponding changes to the text.

1.Although the results seems interesting, they did not give a clear presentation for their results, particularly for the derivation of equations.

All the comments for the eq.(1) describing diffusion of the gas to the growing bubble with a moving boundary, as well as explanation of the boundary conditions, are present in section 2.1. The concept of the excluded volume and appearance of the distribution function are commented step-by-step in sections 2.2 and 2.3.

2.Moreover, there are many results and theories for bubble nucleation in statistical thermodynamics and molecular simulations. The authors seem rarely mentioned those results. They should comment and compare their theories and results with the relevant references. 

We agree that there are many interesting theoretical and simulation results related to the bubble nucleation. This is an old but still actively developed field of research. However, we focus in this paper on the strong influence of the viscosity on the distribution function of the nucleated and growing gas bubbles with non-stationary diffusion where the results were absent. To make it more clear, we added on lines 54-63...The novelty of this task lies in the fact that earlier such calculations of the strong influence of the viscosity on the distribution function of the nucleated and growing gas bubbles with non-stationary diffusion were not carried out at all because of the lack of an analytical basis. Cooperation of the diffusivity and viscosity will be checked for the first time...Earlier, the collective characteristics of an ensemble of gas bubbles were considered only for the case of stationary diffusion in the absence of viscosity...

We also include on lines 368-372 a note about and a reference to the possible development of the theoretical approach presented in this article in connection with recent studies on related topics by other authors....In this paper, we considered the case of instantaneous decompression of a gas-saturated solution. In the case of a finite decompression rate, the approach proposed here requires modification, primarily related to the change in the growth dynamics of a single gas bubble. The corresponding problem of single bubble growth in a highly viscous liquid was considered in recently published paper by Chernov, Davydov, and Pil’nik [32]...

Reviewer 3 Report

The manuscript was evaluated the viscosity and capillarity on Nonequilibrium distribution of gas bubbles in swelling liquid-gas solution. It was well organized and it is required to rechecked all the equations. There are some mistypos in the equations.

Author Response

We thank the referee for his work and valuable comments. Following these comments, we rechecked all the equations in the manuscript. In particular, we corrected indices ex from italic on pp. 4-5 and Fig.1, set tilda above t in eq.(21) and added units for nucleation rate I0 on lines 221, 293, 315 and 320. Unfortunately, some shifts of units and equation numbers provided by reformatting of the original text at submission we were unable to correct.